# Training-Free Multimodal Large Language Model Orchestration

**Tianyu Xie** [1 2] **Yuexiao Ma** [1 2] **Yuhang Wu** [1 2] **Wang Chen** [1 2] **Jiayi Ji** [1 2] **Tat-Seng Chua** [3] **Xiawu Zheng** [1 2 4]
**Rongrong Ji** [1 2 5 6]

## Abstract

Building interactive omni-modal assistants often relies on end-to-end multimodal alignment to fuse heterogeneous modalities, which incurs substantial data and compute costs and limits extensibility. We present Training-Free Large Language Model Orchestration (LLM Orchestration), a training-free *orchestration* framework that integrates off-the-shelf modality experts into a unified multimodal input–output system without additional gradient-based training for integration. LLM Orchestration comprises three components: (1) an LLM controller that infers user intent and emits explicit control tokens for expert selection and sequencing, enabling protocol-constrained and auditable routing; (2) a text-centric cross-modal memory that compresses multimodal evidence into structured records for lightweight retrieval and reuse, reducing redundant expert invocations across turns; and (3) a unified interaction layer that executes routing and memory decisions to support consistent modality transitions, full-duplex streaming, and interruption-aware dialogue. Across diverse multimodal benchmarks, LLM Orchestration achieves strong performance under standard evaluation constraints while maintaining low orchestration overhead and modular upgradeability, providing a practical alternative to costly joint training for omni-modal systems.

---

[1]Media Analytics and Computing Lab, Xiamen University, Xiamen, China [2]Institute of Artificial Intelligence, Xiamen University, Xiamen, China [3]Department of Computer Science, National University of Singapore, Singapore [4]Peng Cheng Laboratory, Shenzhen, China [5]School of Informatics, Xiamen University, Xiamen, China [6]Sino-Russian Research Center for Digital Economy, Xiamen University, Xiamen, China. Correspondence to: Xiawu Zheng <zhengxiawu@xmu.edu.cn>.

*Proceedings of the 43$^{rd}$ International Conference on Machine Learning*, Seoul, South Korea. PMLR 306, 2026. Copyright 2026 by the author(s).

## 1. Introduction

Recent advances in Large Language Models (LLMs) (OpenAI et al., 2023; Team et al., 2023; Grattafiori et al., 2024; Liu et al., 2024a) have enabled strong multimodal capabilities. GPT-4o (Hurst et al., 2024) further demonstrated the feasibility of processing and generating across multiple modalities within a single assistant, motivating increasing interest in omni-modal systems. Conceptually, omni-modal assistants integrate visual, auditory, and textual evidence to support natural interaction and comprehensive understanding (Zhang et al., 2023; Chen et al., 2024b; Wang et al., 2024b; Tong et al., 2024; Fu et al., 2024b). In this paper, we focus on a real-time omni-modal assistant that supports image/video, audio/speech, and text understanding and generation within multi-turn dialogues. A key open question is: how can we build a real-time omni-modal assistant that is *responsive, controllable, and maintainable* as modalities and components evolve?

Despite this progress, many recent multimodal assistants—from vision-language models to emerging omni-modal systems (Zhu et al., 2023; Liu et al., 2023; 2024a; Shin et al., 2024; Alayrac et al., 2022; Fang et al., 2024; Team et al., 2023; Grattafiori et al., 2024; Zhao et al., 2024)—largely follow two integration paradigms. **(A) Unified end-to-end alignment:** a single model is jointly trained or instruction-tuned (often with preference alignment) to handle multiple modalities in one parameter space (Team et al., 2023; Zhao et al., 2024). **(B) Backbone expansion with modality modules:** a base LLM is augmented with modality-specific encoders/adapters/projection layers (e.g., image/video/audio modules) and aligned via paired data (Zhu et al., 2023; Liu et al., 2023; Alayrac et al., 2022; Fang et al., 2024). While effective, both paradigms face two practical limitations. *Cost:* aligning heterogeneous modalities typically requires curated datasets and intensive fine-tuning, incurring substantial human effort and compute. *Rigidity:* component upgrades or modality additions often trigger retraining (e.g., swapping an ASR front-end, upgrading a VLM, or adding a new video expert), slowing iteration and complicating deployment. Meanwhile, purely ad-hoc tool-use pipelines often lack system-level guarantees on verification, traceability, and interruption consistency, making them hard to measure and maintain in real-time settings.

Beyond these two paradigms, real-time omni-modal assistants raise a complementary challenge that is not primarily about *representation learning*, but about *execution and interaction*. In multi-turn, streaming dialogues, the system must make online decisions under strict latency constraints: which computations to trigger, how to manage intermediate states across modalities, and how to behave under user barge-in and partial outputs. These requirements are orthogonal to whether the underlying capability is obtained via unified alignment or modular encoders, yet they critically affect responsiveness, controllability, and maintainability in deployment. This suggests that, in addition to model integration, we need an explicit orchestration abstraction with well-defined runtime behaviors and measurable guarantees.

To address these limitations, we propose a **training-free orchestration** framework that composes specialized experts into a single real-time omni-modal assistant. Our key idea is simple: an off-the-shelf LLM controller emits *schema-constrained system tokens* that (i) route each turn to appropriate modality experts and (ii) support interaction control (e.g., interruption), while a text-centric memory *selectively reuses* previously generated multimodal evidence under protocol-enforced rules. Concretely, when the executor verifies that the current turn matches a committed evidence key (e.g., identical image hash / segment id) and reuse is permitted by runtime policies, it retrieves the corresponding cached record instead of re-invoking the expensive modality expert, thereby reducing *unnecessary* expert calls without relying on ambiguous semantic similarity. **Training-free definition:** throughout this work, "training-free" means we perform *no gradient-based training or fine-tuning* for orchestration components (controller, routing logic, memory, and interaction manager); we only use static prompt specifications (token-to-expert mapping and token schema) to integrate off-the-shelf experts. We do not claim training-free multimodal capability; rather, we claim training-free *integration and control* without additional alignment, instruction tuning, or preference optimization for integration.

**Positioning.** Our system composes external experts but differs from generic tool-use agents by defining a *system-level protocol*: closed-vocabulary control tokens coupled with a static token-to-expert mapping (interfaces and I/O contracts). Execution is *router-enforced*: tokens are validated and resolved under explicit runtime policies (time-outs/cancellation), and violations are rejected with fallback routing. The protocol yields a replayable routing trace (token → route decision → cache-or-call → output) and interruption-consistent interaction: on barge-in, in-flight jobs are canceled, ephemeral buffers are separated from committed memory, and only finalized segments/results are persisted.

Our framework comprises three tightly coupled components: **(C1) Controller-based orchestration**—a central controller emits explicit, auditable tokens to assign each turn to experts; **(C2) Cross-modal memory integration**—a text-centric memory compresses and retrieves multimodal evidence to preserve context and reduce redundant expert calls; and **(C3) Unified interaction layer**—a system-level manager executes routing and memory decisions to provide streaming omni-modal I/O with interruption handling. Together, these components offer an interpretable, modular, and extensible alternative to end-to-end joint training.

The main contributions are:

- **(C1) Structured control-token routing.** We introduce an LLM controller that selects and sequences specialized experts via explicit system tokens (e.g., *[S.need_vision]*, *[S.need_audio]*, *[S.stop]*), grounded in a fixed token-to-expert mapping and I/O contracts. We report routing latency/overhead and analyze its sensitivity across different expert latency regimes (Figure 2). We further demonstrate extensibility via expert swapping under a fixed controller (Table 2).

- **(C2) Lightweight cross-modal memory compression and reuse.** We propose a text-centric structured memory that stores expert outputs as compact records and retrieves them using lexical matching with recency signals for bounded worst-case latency and explainable hits. Redundant expert calls are defined by exact evidence-key consistency (e.g., image hash, video chunk id, audio segment timestamp), which avoids ambiguous semantic matching and reduces false reuse. We quantify redundant-call reduction and the resulting savings in Figure 2.

- **(C3) Unified interaction layer for real-time omni-modal I/O.** We design an interaction manager that supports full-duplex streaming and interruption-aware control, where *[S.stop]* cancels in-flight expert/TTS execution and triggers re-routing under updated intent. We evaluate interruption handling with a reproducible test protocol and report cancellation behavior, wasted computation, and end-to-end latency impacts in Section 4.

Under standard benchmark constraints and prompts, our system achieves 44.07% on WorldSense (Hong et al., 2025) and 69.37% on MMStar (Chen et al., 2024a). Compared with end-to-end multimodal baselines and our ablated variants (e.g., without memory reuse or protocol-constrained routing), we maintain strong accuracy while keeping routing overhead below 12% of end-to-end latency and reducing redundant expert invocations in multi-turn interactions (Section 4). We follow official evaluation protocols and

scoring scripts, and use experts strictly for modality perception/format conversion without external knowledge access. Detailed accuracy, ablations, and latency breakdowns are provided in Section 4.

## 2. Related Work

### 2.1. Training-Based Omni-Modal Systems

Multimodal Large Language Models have advanced rapidly via end-to-end multimodal alignment, using either full-parameter training or parameter-efficient adaptation. VITA (Fu et al., 2024b) adopts multi-stage instruction tuning and modality alignment to improve cross-modal coherence. To reduce training cost, parameter-efficient approaches freeze the LLM backbone and train lightweight modality components, e.g., Freeze-Omni (Wang et al., 2024d), Mini-Omni2 (Xie & Wu, 2024), LLaMA-Omni (Fang et al., 2024), and Moshi (Défossez et al., 2024). Recent real-time spoken assistants further incorporate streaming speech understanding and generation, e.g., LLaMA-Omni2 (Fang et al., 2025).

A recurring limitation is that integration is coupled to specific modality interfaces (e.g., encoders/adapters/projection layers) and paired multimodal data, so component upgrades (e.g., swapping ASR or vision modules) can require additional alignment to preserve calibration and coherence. In contrast, we study *training-free integration and control*: rather than proposing a new omni model, we provide an *orchestration/integration layer* over off-the-shelf experts.

### 2.2. Training-Free Routing and Model Composition

Training-free orchestration composes specialized models via an LLM planner/controller and a tool/model interface. HuggingGPT (Shen et al., 2023) uses natural-language tool descriptions with multi-step planning. Visual Programming (Gupta & Kembhavi, 2023) and ViperGPT (Surís et al., 2023) compose visual tools via program synthesis, while ReAct (Yao et al., 2023) interleaves reasoning and action for tool invocation. Chen et al. (Chen et al., 2023) show that careful prompting can enable training-free coordination of visual modules. Recent work also studies multimodal tool agents that learn to invoke tools under multimodal observations, e.g., MLLM-Tool (Wang et al., 2025).

A related line improves the reliability of tool invocation by enforcing structured outputs, either via constrained generation engines (e.g., grammar-/schema-based decoding) (Dong et al., 2025) or by learning schema-following policies with supervision or reinforcement learning (Lu et al., 2025). These works motivate *parseable, verifiable* action representations, consistent with our closed-vocabulary tokens and router-side validation.

Many prior orchestration systems operate in action spaces that are often open-ended (natural language, code, or free-form arguments), which complicates validation and

auditing in low-latency interactive settings. We instead introduce a *protocol-constrained* control-token interface grounded in a static token-to-expert mapping with explicit I/O contracts (Section 3.3); tokens are validated and executed under explicit runtime policies (e.g., timeouts/cancellation). This yields a replayable routing trace (token → route decision → cache-or-call → output), supports measurable routing overhead (Figure 2), and enables explicit rejection of invalid tokens or contract violations.

### 2.3. Cross-Modal Memory for Efficient Multi-Turn Orchestration

Memory mechanisms are widely used in interactive assistants to preserve context and cache intermediate results across turns. Recent work studies how memory structure and retrieval affect agent behaviors and robustness (Zeng et al., 2024; Xu et al., 2025b). In multimodal orchestration, reuse decisions additionally need to be *verifiable* across modalities and *latency-bounded* for real-time interaction.

We adopt a text-centric, structured cross-modal memory that records expert outputs as compact, reusable records and retrieves them with lightweight rules and recency signals (Section 3.3). Unlike long-horizon persona/preference memory, we focus on caching modality evidence for multi-turn perception and reasoning. We prioritize *evidence-key consistency* (e.g., image hash, video chunk id, audio segment timestamp) to determine reuse eligibility, favoring verifiability over semantic similarity and mitigating erroneous reuse. We operationalize redundant expert invocations and quantify call reduction via ablations (Figure 2).

### 2.4. Interactive Omni-Modal Assistants

Interactive omni-modal assistants emphasize streaming responsiveness and robustness to user interruptions beyond offline accuracy. VITA (Fu et al., 2024b) studies interruption handling in an end-to-end training setting, while HumanOmni (Zhao et al., 2024) targets human-centric interaction scenarios. Recent spoken assistants further investigate real-time speech understanding/generation with barge-in and interruption handling, e.g., SALMONN-omni (Yu et al., 2024) and LLaMA-Omni2 (Fang et al., 2025).

Our work is complementary: rather than learning interaction behaviors end-to-end, we provide a unified interaction layer that *operationalizes* routing and memory decisions into explicit execution semantics for real-time pipelines, including canceling in-flight expert/TTS execution and separating ephemeral buffers from committed memory under clear commit rules (Section 3.4).

### 2.5. Multi-Agent Collaboration Frameworks

Multi-agent frameworks distribute tasks across multiple interacting LLMs and specialized roles. AutoGen (Wu et al., 2023) proposes flexible dialogue patterns for task decomposition. mmctagent (Kumar et al., 2024) en-

hances multimodal decision-making via multi-agent co-ordination, while CrewAI (Duan & Wang, 2024) and TaskWeaver (Qiao et al., 2023) focus on role-based orchestration and workflow automation. Domain-specific systems such as LawLuo (Sun et al., 2024) simulate professional consultations, and Self-Organized Agents (Ishibashi & Nishimura, 2024) and CMAT (Liang et al., 2024) explore coordination for code generation and small-model enhancement.

While multi-agent designs can improve flexibility for complex planning, they may incur extra communication overhead and latency variance due to negotiation, which is undesirable in low-latency interactive settings. Our architecture instead adopts a single-controller protocol augmented with cross-modal memory, reducing coordination overhead and latency variance while retaining explicit and auditable routing decisions (Sections 3.3–3.3).

These observations motivate an orchestration abstraction that makes *routing*, *reuse*, and *interruption handling* explicit and measurable while remaining modular to component evolution. Our work takes a step toward this goal by making orchestration and interaction control explicit and measurable in a training-free integration layer over off-the-shelf experts.

## 3. Method

### 3.1. Motivation and Design Goal

Recent omni-modal assistants achieve strong interactive capabilities by aligning audio, vision, and language through substantial training (e.g., VITA (Fu et al., 2024b)).[1] While effective, alignment-centric designs are costly and can tightly couple components, making upgrades (e.g., swapping ASR/LLM/TTS) difficult without re-tuning.

We ask whether omni-style real-time interaction can be achieved with off-the-shelf experts and explicit runtime control, rather than requiring a newly trained omni model. Our key observation is that real-time omni behavior hinges on *interaction-time control*—when to listen vs. respond, when to stop on user barge-in, and which modality evidence to acquire—beyond representation learning alone.

We therefore target **training-free integration and control**: the orchestration layer (routing, memory reuse, streaming, and interruption semantics) is specified by a fixed protocol and deterministic execution rules, with no gradient-based training for integration components. Concretely, an LLM controller emits closed-vocabulary state/modality to-

kens that a deterministic router executes to invoke or reuse pluggable experts, while an evidence-keyed memory enables verifiable cache-or-call reuse across turns. Our objective is to match omni interaction behavior through modular orchestration rather than learned multimodal alignment.

### 3.2. Stage 1: Unified Input

Stage 1 runs continuously and normalizes heterogeneous inputs into two outputs for the next stage: a consolidated query $q_t$ and a set of evidence references $\mathcal{K}_t$. For speech, Voice Activity Detection (VAD) segments the microphone stream into intervals and ASR transcribes each interval into text tokens. We denote the resulting transcript evidence as $text_{audio}$; Stage 1 buffers/merges these tokens and consolidates them into $q_t$ (with timestamps retained for bookkeeping).

For non-text inputs, Stage 1 does not expose raw payloads to the controller; instead it assigns each segment an *evidence key* using system code (KEYGEN) and returns the keys as references. An evidence key is a system-generated identifier that uniquely names a concrete multimodal segment; Stage 2 refers to raw inputs only through these keys and never directly accesses the payload:

$$k = \text{KEYGEN}(m, s) \triangleq \langle m,\ s,\ h \rangle, \tag{1}$$

where $m$ is the modality tag, $s$ is a segment identifier (audio timestamp span, image fingerprint, or video chunk/frame id), and $h$ is an integrity checksum. Stage 1 maintains a local mapping from $k$ to the raw payload pointer; the controller only observes $\mathcal{K}_t$.

For images/videos, Stage 1 always registers segments and outputs their keys, but vision-to-text evidence is computed only on demand: the vision expert is silent by default and is invoked only after Stage 2 requests vision (e.g., vision); upon activation, Stage 1 produces a textualized description $text_{vision}$ for the requested key(s). During system playback, any new valid VAD segment is flagged as a barge-in event and reported to Stage 2 for decision making.

### 3.3. Stage 2: LLM Orchestration

Stage 2 is the decision and execution core. At turn $t$, it takes $(q_t, \mathcal{K}_t)$ from Stage 1, together with interaction events (e.g., barge-in), and a bounded memory view $\hat{M}_{t-1}$ derived from its own session memory $M_{t-1}$. Stage 2 then produces either a control decision (keep listening / stop output) or a response text $o_t^{text}$ prepared for streaming.

**Controller → Router → Experts.** We use an off-the-shelf LLM as a controller. Given a fixed system prompt that specifies the closed token vocabulary and the current interaction state, the controller emits a short sequence of control tokens indicating (i) system status (listen/stop/respond) and (ii) which modality evidence is required (e.g., vision for some referenced key). A deterministic router validates

---

[1]A common abstraction for such systems is gradient-based multimodal alignment on paired observations, e.g., $\theta^{\star} = \arg\min_{\theta} \mathbb{E}_{(a,v,y)\sim\mathcal{D}} \left[ \mathcal{L}_{align}(f_{\theta}^{a}(a), f_{\theta}^{v}(v), f_{\theta}^{y}(y)) \right]$, where $a$ and $v$ denote audio and visual inputs, $y$ denotes language supervision, and $\mathcal{L}_{align}$ enforces cross-modal consistency under an end-to-end or staged recipe.

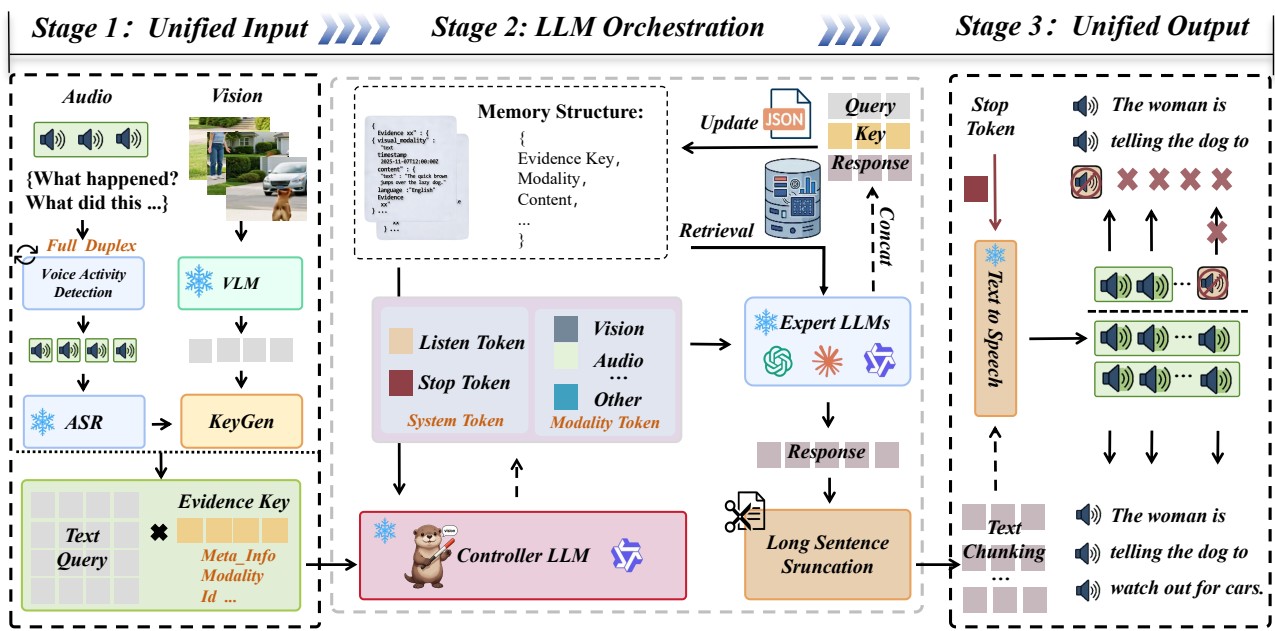

Figure 1. Overview of our training-free MLLM Orchestration pipeline. Stage 1 normalizes multimodal inputs into a unified representation with evidence keys. Stage 2 uses a controller to emit closed-vocabulary control tokens, routes to external experts, and performs cache-or-call retrieval with evidence-keyed memory. Stage 3 streams ordered outputs and supports barge-in by canceling pending jobs.

these tokens and resolves them through a system-side mapping $\mathcal{E}$, which binds each routing token to an expert endpoint, required inputs, output schema, and runtime policies (timeouts/cancellability). This step selects the concrete expert(s) to invoke and constructs their inputs by combining (a) the user-side query $q_t$, (b) the referenced evidence keys in $\mathcal{K}_t$ (used to locate raw payloads outside the controller), and (c) controller-organized textual context from $\hat{M}_{t-1}$ when needed. Each invoked expert then returns textualized evidence (as text tokens) that is schema-validated by the router before being used downstream.

**Evidence-keyed memory and cache-or-call.** To reduce redundant expert invocations across turns, Stage 2 maintains an evidence-keyed memory $M$ as a per-session store of *textualized evidence records* indexed by evidence key $k$. A record contains a compact, schema-valid summary of an expert output (e.g., ASR transcripts $text_{\text{audio}}$, vision descriptions $text_{\text{vision}}$), together with lightweight metadata (modality tag, timestamps). Memory stores *evidence*, not raw payloads: raw audio/images/videos remain outside the controller and are accessible only to experts via evidence references. To bound controller context length, Stage 2 exposes only a compressed view $\hat{M}_{t-1} = \text{COMPRESS}(M_{t-1})$ in the prompt.

When evidence is required (e.g., vision for $k \in \mathcal{K}_t$),

Stage 2 applies the deterministic rule:

$$d(k) = \begin{cases} \text{READ}(M_{t-1}, k), & \text{if } k \in M_{t-1}, \\ \text{CALLEXPERT}(k) \text{ and write to } M_t, & \text{otherwise.} \end{cases} \quad (2)$$

Key equality is the default reuse criterion, making reuse decisions verifiable. In particular, vision textualization is gated by control tokens: when vision is present, the selected vision expert is invoked on the referenced key(s) and produces $text_{\text{vision}}$; if the corresponding record already exists in memory, the expert call is skipped. Analogously, when need_audio is present, the ASR/audio expert produces (or reuses) $text_{\text{audio}}$.

**Response synthesis and streaming preparation.** After evidence acquisition, Stage 2 concatenates (i) the consolidated query $q_t$, (ii) retrieved memory records relevant to $\mathcal{K}_t$ (via $\hat{M}_{t-1}$), and (iii) any newly produced expert evidence into a single textual context (Concat in Figure 1), applies a lightweight truncation/compression rule to bound context length, and generates the response text $o_t^{\text{text}}$ with a text-only generator. To prepare for real-time output, Stage 2 further segments $o_t^{\text{text}}$ into speakable chunks using punctuation and a maximum-token constraint; these chunks (with monotonic chunk IDs) are then passed to Stage 3 for parallel Text-to-Speech (TTS) and ordered playback. Finally, Stage 2 commits the response and the evidence keys used as a structured session record in $M_t$ and logs a replayable trace (tokens → routes → cache-or-call → outputs).

### 3.4. Stage 3: Streaming Output with Barge-in Cancellation

Stage 3 takes the response text $o_t^{\text{text}}$ and control decisions (e.g., `stop`) from Stage 2 and realizes them as ordered streaming I/O. When a response is to be spoken, Stage 3 chunks $o_t^{\text{text}}$ by punctuation and a maximum length, assigns monotonic chunk IDs, and synthesizes chunks asynchronously with a fixed-size worker pool. Chunks may finish out of order, but playback is strictly ordered by chunk ID.

Interruption is enforced by cancellation. When Stage 2 emits `stop` (e.g., after receiving a barge-in event), Stage 3 cancels pending/running TTS jobs and clears both the generation queue and playback buffer, guaranteeing silence after interruption. We apply the same cancellation policy to expert calls marked cancellable in $\mathcal{E}$, ensuring consistent interruption behavior across perception and rendering.

### 3.5. Multi-turn Full-Duplex Workflow

Algorithm 2 summarizes the multi-turn interaction loop, including barge-in interruption, deterministic planning, evidence acquisition via cache-or-call, streaming TTS, and commit-on-complete memory updates. We provide full workflow pseudocode in Appendix E for readability.

### 3.6. Pipeline Overview

**Running example (Figure 1).** Consider a user who speaks "What happened?" while providing a short video clip where a dog runs as a car approaches. In Stage 1, VAD+ASR transcribes the utterance into $text_{\text{audio}}$ and consolidates it into the query $q_t$; the video is registered with an evidence key $k_{\text{vid}} \in \mathcal{K}_t$ via KEYGEN (without running vision by default). In Stage 2, the controller, conditioned on the system prompt and $(q_t, \mathcal{K}_t, \hat{M}_{t-1})$, emits `respond` with `vision` for $k_{\text{vid}}$; the router resolves this token to the vision expert, applies cache-or-call on $k_{\text{vid}}$, and obtains (or reuses) a textualized description $text_{\text{vision}}$ (e.g., the dog notices the car and moves away). The generator then produces the response text (e.g., "The woman is telling the dog to watch out for cars.") and Stage 2 segments it into speakable chunks for Stage 3. Finally, Stage 3 performs parallel Text-to-Speech (TTS) on the chunks and plays them in order; if the user barges in mid-playback, Stage 2 emits `stop` and Stage 3 cancels pending jobs and clears buffers.

## 4. Experiments and Results

We evaluate our training-free orchestration along three system-level claims: (C1) protocol-constrained routing for robust control and modular composition, (C2) evidence-keyed memory for reuse and efficiency, and (C3) an interaction layer for streaming and interruption-consistent runtime behavior. We report end-to-end benchmark results in Table 1 and isolate each claim via controller/expert swap-ping, memory ablations, and compute-only latency break-downs (Figure 2, Table 2).

### 4.1. Experimental Setup

**Controller & Expert Configuration.** We employ Qwen2.5-14B (Chu et al., 2024) as the central controller LLM with deterministic decoding (temperature=0.0, top-p=1.0) to ensure stable control-token emission. The system exposes a set of *pluggable* experts accessed via a system-side token-to-expert mapping. The controller emits closed-vocabulary state/modality tokens (e.g., `need_vision`, `listen`, `stop`); experts are invoked only when routed by these tokens. When no expert is required, the controller performs text-only reasoning. In our Video-MME pipeline, we primarily use frame-based visual experts (e.g., Qwen2.5-VL) under a fixed sampling budget, while dedicated video experts (e.g., LLaVA-Video) can be plugged in when temporal modeling is desired. Unless otherwise stated, the reported Video-MME results use frame-based experts plus ASR transcripts (Section 4.3). All local experts use greedy decoding (temperature=0.0) unless otherwise noted. We run inference on 8×NVIDIA A100 GPUs (80GB) with mixed precision (FP16).

**Baselines.** We compare against: (i) *single-model open-weight MLLMs* (e.g., Qwen2.5-VL, InternVL-2, LLaVA-OV), (ii) *jointly trained omni-models* (e.g., Qwen2.5-Omni (Xu et al., 2025a), VITA (Fu et al., 2024b), M2-omni (Guo et al., 2025)), and (iii) *commercial APIs* (GPT-4o (Hurst et al., 2024), Claude 3.5 (Anthropic, 2024), Gemini-1.5-Pro (Team et al., 2024)) as reference points. For our system, Table 1 reports the average active visual expert parameters under dynamic routing; the controller is fixed at 14B. *Training-free* means we perform no additional joint end-to-end training or cross-modal alignment to obtain the unified assistant; experts are off-the-shelf pre-trained/aligned models coordinated via prompts and explicit control tokens.

**Evaluation Protocol.** We evaluate on established benchmarks covering: general multimodal understanding (MME (Fu et al., 2023), MMBench-EN/CN (Liu et al., 2024b)), challenging vision QA (MMStar (Chen et al., 2024a), MMMU (Yue et al., 2024)), long video understanding (LVBench (Wang et al., 2024c), Video-MME (Fu et al., 2024a)), holistic omni-modal evaluation (World-Sense (Hong et al., 2025)), and specialized domains (Math-Vision (Wang et al., 2024a), CC-OCR (Yang et al., 2024)). We follow official evaluation scripts when provided and fix the random seed (seed=42). For video benchmarks, we standardize frame sampling to FPS=20 and cap processing to *MAX_FRAMES_PER_BATCH*=20 unless a benchmark protocol constrains otherwise.

**Latency Measurement.** We report *compute-only* latency for local models, including controller inference, token pars-

ing/validation, memory lookup, scheduling decisions, and expert inference, excluding network I/O. Timing statistics are computed over 100 randomly sampled queries per benchmark; we additionally report system transmission overhead (e.g., IPC) separately when relevant.

## 4.2. Comparison with Mainstream Omni Models

This subsection reports end-to-end comparisons against mainstream multimodal/omni models. Table 1 summarizes unified results across general multimodal, vision, and video benchmarks.

**Main Results (Table 1).** Table 1 summarizes end-to-end performance across representative multimodal, vision, long-video, and holistic omni settings. Overall, our orchestration is competitive with strong open-weight baselines and approaches natively trained omni models in aggregate. We attribute the gains to protocol-driven expert routing and explicit evidence composition (e.g., combining visual evidence with ASR transcripts), without requiring joint end-to-end training for integration.

**Efficiency (Avg. Inference Latency).** For methods with reported timing, latency reflects the usual accuracy–speed trade-off. Our system operates in the same order of magnitude as fast open-weight baselines, while preserving competitive accuracy (Table 1). Detailed latency decomposition and overhead are reported in Figure 2.

> **Finding A (Competitive end-to-end performance).**
> Across representative benchmarks, protocol-driven orchestration with off-the-shelf experts achieves competitive end-to-end accuracy, indicating that strong omni-modal capability can be obtained without joint end-to-end training for integration (Table 1).

**(C1) Controller Replacement under a Fixed Token Protocol.** Under the same token protocol, prompts, deterministic decoding, and runtime semantics, we replace the controller while keeping the expert routing and execution unchanged. The resulting accuracy is similar, suggesting routing behavior is primarily constrained by the protocol-defined action space rather than a specific controller implementation.

**(C2) Memory Integration Efficiency.** We ablate memory by disabling retrieval and forcing recomputation at each turn under the same protocol and runtime semantics. Memory substantially reduces redundant expert invocations, with larger gains on sustained-query dialogues where evidence is repeatedly referenced (Figure 2).

> **Finding B (Modularity and efficiency with low overhead).** Under a fixed token protocol and runtime semantics, controller/expert swapping preserves behavior while accuracy scales with expert capability

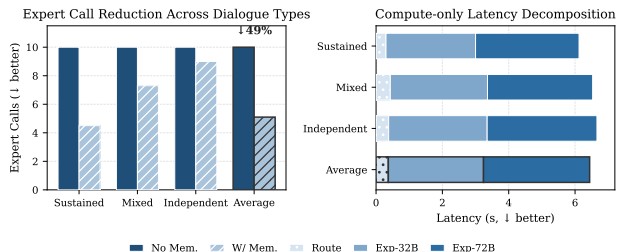

*Figure 2.* Efficiency analysis. Left: expert-call reduction across dialogue types when enabling cross-modal memory (avg. 49%). Right: compute-only latency decomposition; expert inference dominates while routing and orchestration overhead remain small. Here, $T_{route}$ includes controller inference, token parsing/validation, memory lookup (when enabled), and scheduling, and the same measurement scope is used across settings for a fair overhead comparison. Overhead is computed as $100 \times \frac{T_{route}}{T_{route}+T_{Exp-72B}}$ and stays below 12% across scenarios (avg. 10.36%).

> (Table 2). Evidence-keyed memory reduces redundant expert calls (Figure 2), and routing/scheduling overhead remains a small fraction of total compute.

**(C1/C3) Orchestration Overhead Analysis.** Figure 2 shows that expert forward passes dominate compute-only latency, while routing (controller inference, token parsing, memory lookup, scheduling) adds limited overhead. We additionally measure a small transmission overhead (excluded from compute-only time) and report it separately.

## 4.3. Video Understanding Performance and Holistic Evaluation

We evaluate audio-visual understanding under a standardized frame-sampling budget with ASR transcripts as textual audio evidence. Protocol-driven routing enables selective expert invocation and explicit evidence aggregation, which is particularly helpful as temporal context grows under sparse sampling. Stratified results by video length are summarized in Figure 3.

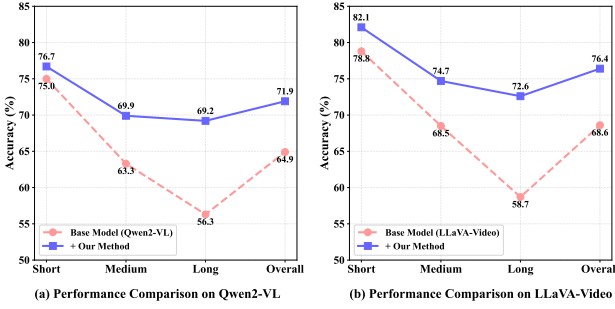

*Figure 3.* Video-MME accuracy stratified by video length. Our orchestration consistently improves over the corresponding single-model baseline under the same backbone, with larger gains on longer videos, indicating that protocol-driven routing and evidence aggregation are particularly beneficial when temporal context grows. (a) Qwen2-VL backbone. (b) LLaVA-Video backbone.

| Model | Avg. Active Params | General Multimodal | | | Vision Understanding | | | | Efficiency |
|---|---|---|---|---|---|---|---|---|---|
| | (B) | MME | MMBench-EN | MMStar | LVBench | MMMU | Video-MME | WorldSense | Time (s) |
| Qwen2.5-VL | 7 | 1673 | 84.45 | 59.94 | 45.30 | – | 56.62 | – | – |
| Qwen2.5-VL | 32 | 1915 | 85.55 | 66.43 | 49.00 | – | 62.39 | – | – |
| Qwen2.5-VL | 72 | 1980 | 86.61 | 68.22 | 47.30 | – | 65.74 | – | – |
| Qwen-VL-Max | - | 2281 | 77.60 | – | – | – | 51.30 | – | – |
| Qwen2.5-Omni | 7 | **2340** | 81.80 | 64.00 | – | 59.20 | 64.30 | 45.4 | 6.0 |
| VITA1.5 | 7 | 2006 | 71.80 | 46.40 | – | 47.30 | 59.20 | 36.9 | 3.7 |
| LLaVA-OV | 7 | – | 80.80 | 61.70 | – | – | 58.20 | 37.7 | – |
| LLaVA-OV | 72 | – | 85.90 | 66.10 | 26.90 | – | 66.20 | – | – |
| InternVL-2 | 8 | – | 81.70 | 59.40 | – | – | – | 39.1 | – |
| InternVL-2 | 26 | – | 83.40 | 60.40 | – | – | – | – | – |
| IXC2.5 | 7 | – | – | – | – | – | 60.60 | – | – |
| M2-omni | 9 | – | – | 60.50 | – | 51.20 | 60.40 | – | – |
| Gemini-1.5-Pro | - | – | – | – | 33.10 | – | **75.00** | 48.0 | – |
| GPT-4V | - | 1409 | 75.00 | 57.10 | – | – | 59.90 | – | – |
| GPT-4o | - | 2310 | 83.10 | 64.70 | 34.70 | 59.20 | 71.90 | 42.6 | 1.2 |
| **Ours:base Qwen2.5** | 53.8 | 1922 | **88.54** | **69.37** | **50.27** | **70.04** | 65.58 | 44.1 | 3.2 |

*Table 1.* Unified comparison on general multimodal and vision/video benchmarks. **Avg. Active Params (B)** reports dense model size for single-model baselines and the average active visual expert budget for our routed system; our controller is fixed at 14B and not included in this column. "–" indicates unavailable or undisclosed information. **WorldSense** measures holistic omni-modal understanding. Best/second-best marked by **/***. Latency and overhead analyses are reported in Figure 2.*

*Table 2.* Video-MME under a fixed controller: expert-pair swapping (20 FPS, *MAX_FRAMES_PER_BATCH*=20).

| Expert pair | Acc (%) | Usage (V/R, %) |
|---|---|---|
| **V:** Qwen2.5-VL-32B
**R:** Qwen2.5-VL-72B | 65.58 | 45.62/54.38 |
| **V:** Qwen3-VL-32B
**R:** Qwen3-VL-235B-A22B | 72.31 | 43.49/56.51 |

**Expert Swapping Ablation (Video-MME).** To verify extensibility, we keep the controller fixed and swap the frame-based visual and reasoning experts under identical evaluation settings. With Qwen2.5-VL-32B + Qwen2.5-VL-72B, we obtain **65.58%**. Replacing them with Qwen3-VL-32B-Instruct + Qwen3-VL-235B-A22B-Instruct (Yang et al., 2025) improves accuracy to **72.31%** (same ASR pipeline and frame sampling: 20 FPS. This ablation demonstrates expert extensibility: keeping the controller and protocol fixed, upgrading the expert pair improves accuracy while preserving the same orchestration semantics (Table 2).

We further evaluate holistic omni-modal capability on WorldSense (Hong et al., 2025). On WorldSense, our orchestration remains competitive under a holistic omni setting (Table 1). The task-type breakdown (Appendix F, Table 8) suggests a temporal-resolution trade-off under sparse sampling: high-level semantic/causal reasoning benefits from modular evidence integration, while fine-grained temporal counting remains challenging.

**Limitations and Trade-offs.** Fine-grained temporal counting remains challenging with sparse frame sampling. This reflects an accuracy–resolution trade-off: modular evidence helps high-level reasoning, whereas dense temporal encoders better capture fast dynamics.

## 5. Conclusions

We presented a training-free LLM orchestration framework that composes off-the-shelf multimodal experts into a unified full-duplex assistant through a protocol-constrained control interface. A central controller emits closed-vocabulary state/modality tokens that are deterministically executed by the runtime router, enabling auditable expert invocation, interruption-consistent streaming (e.g., cancellation and safe fallbacks), and modular extensibility without joint end-to-end training for integration. We further introduced an evidence-keyed, text-centric cross-modal memory that supports multi-turn reuse via cache-or-call and bounded compression, improving efficiency while keeping orchestration overhead small. Experiments across heterogeneous benchmarks and system ablations show that the protocol largely decouples orchestration from expert implementations, so controller/expert upgrades can translate into capability gains without retraining.

**Future work.** Current limitations include sparse temporal sampling for fine-grained events and non-oracle routing in ambiguous cases. Future work will study higher-resolution evidence retention, specialized temporal experts, and lightweight routing refinement while preserving protocol interpretability. Taken together, this work provides a scalable recipe for building real-time omni-modal assistants from evolving experts while keeping execution semantics explicit.

## Acknowledgements

This work is supported by the National Key Research and Development Program of China (No. 2025YFE0113500),

the National Natural Science Foundation of China ( No. 62576299) and the Fundamental Research Funds for the Central Universities.

## Impact Statement

This paper presents work whose goal is to advance the field of Machine Learning. There are many potential societal consequences of our work, none which we feel must be specifically highlighted here.

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

# Appendix

**Table 3.** Control token vocabulary used in all experiments. Routing tokens trigger mapping-bound expert calls; interaction tokens affect dialogue state and execution control. The pattern *[S.need_*]* is extensible via the mapping without changing the parser or router.

| Token | Type | Router semantics |
|---|---|---|
| *Routing tokens* | | |
| [S.need_vision] | route | invoke vision expert on referenced media segment(s) |
| [S.need_video] | route | invoke video expert on sampled frames / video segment(s) |
| [S.need_audio] | route | invoke ASR/audio expert and return transcript/evidence |
| [S.need_ocr] | route | invoke OCR expert when text extraction is required |
| [S.need_math] | route | invoke math specialist when required by benchmark |
| *Interaction tokens (dialogue control)* | | |
| [S.listen] | interact | pause and await additional user input; keep state |
| [S.stop] | interact | cancel in-flight execution when permitted by policy |
| [S.speak] | interact | allow downstream TTS rendering of the final text response |

## A. Core Instruction Set: Communication Protocol

The controller and router communicate through a closed-vocabulary token protocol. Each routing token is bound to a mapping entry that specifies the target expert, required inputs, output schema, and runtime policies (e.g., timeout, cancellability). Interaction tokens are handled by the interaction manager and do not invoke experts.

**Deterministic validation.** The router validates that the controller output is tokens-only and that all routing tokens are well-formed and mapping-resolvable. Invalid outputs trigger rejection and a safe fallback route that produces a bounded text-only response.

## B. Core Prompt Design: Dialogue Control Framework

We provide the controller with (i) a protocol specification defining the closed token vocabulary, (ii) a mapping describing expert interfaces and runtime policies, and (iii) routing guidelines that encourage evidence reuse when valid keys are available. The generator receives only text evidence (retrieved expert outputs and compact memory view) and is never parsed for control tokens.

**Controller prompt skeleton.**

```
Role:  You are the orchestration
controller.
Input:  (a) user query q_t; (b) compact
memory view M̂_{t-1}; (c) mapping summary.
```

```
Output constraint:  Emit a sequence
of control tokens only, each from the
allowed vocabulary.  Do not output any
other text.
Routing guideline:  Prefer reusing
cached evidence when a matching evidence
key exists; avoid redundant expert
calls.
Failure mode:  Invalid outputs will be
rejected by a strict parser and replaced
by a safe default route.
```

**Mapping snippet (illustrative).**

```
Token [S.need_vision] → Expert:
vision_expert
Inputs:  current image/video frames OR
reusable evidence_key
Output schema:  structured textual
evidence (caption/objects/attributes)
Policy:  timeout T_vision, cancellable=true

Token [S.need_audio] → Expert:
asr_expert
Inputs:  current audio segment OR
reusable evidence_key
Output schema:  transcript + timestamps
Policy:  timeout T_asr, cancellable=true
```

**Generator prompting.** The generator is invoked with a disjoint prompt that contains $(q_t, \widehat{M}_{t-1}, D_t)$ and is required to produce a final text response only. When the same underlying LLM is used for both roles, we use two separate calls with non-overlapping prompts; only the controller call is parsed for control tokens.

## C. Memory Pool Schema: Cross-modal Memory Management

The memory pool stores expert outputs as structured text records indexed by evidence keys. The goal is verifiable evidence reuse under bounded latency, rather than long-horizon persona storage. Persistent writes are restricted to completed expert outputs and finalized segments; partial hypotheses and canceled calls remain ephemeral (Appendix D).

**Record format.** Each memory entry is a tuple:

$$e_j = \{\text{turn}, \text{modality}, \text{evidence\_key}, \text{payload}, \text{meta}\},$$

where *payload* is a compact textual/JSON-like expert output, and *evidence_key* identifies the originating input segment.

**Memory entry examples (illustrative)**

```
# Evidence key
key := (modality, source, segment, checksum)

# Vision entry (abridged)
{
"turn":  3,
"modality":  "vision",
"key":  ["vision","img_042","full","sha1:..."],
"payload":  {"caption":  "..."},
"meta":  {"model":  "Qwen2.5-VL-32B"}
}

# ASR entry (abridged)
{
"turn":  4,
"modality":  "audio",
"key":  ["audio","aud_017","t=12-18","sha1:..."],
"payload":  {"transcript":  "..."},
"meta":  {"model":  "ASR"}
}
```

**Compression view.** To respect context-length limits, we maintain a compact view $\widehat{M}_t$ by selecting a bounded set of records using a weighted score that combines recency, lexical relevance to $q_t$, and modality coverage. The full memory pool $M_t$ is retained for router-side retrieval via evidence keys; the controller and generator read only $\widehat{M}_t$.

## D. Modality-Specific Processing Details

This section documents implementation details for standard input preprocessing and output rendering modules that operate outside the core orchestration logic. These modules handle modality-specific transformations to ensure compatibility with the controller's text-centric interface.

**Input Preprocessing:** Raw multimodal inputs are converted to text-compatible formats via standard modules: ASR (Automatic Speech Recognition) for audio input, OCR (Optical Character Recognition) for text extraction from images, and frame extraction for video preprocessing. These preprocessing steps are orthogonal to orchestration and utilize off-the-shelf tools.

**Output Rendering - TTS Segmentation:** For interactive voice applications, the text response $o_t$ may be synthesized to speech via TTS (Text-to-Speech). The TTS module employs a rule-based segmentation strategy to optimize speech synthesis quality and latency. The segmentation rules are designed to maintain semantic coherence while enabling parallel processing. Below are the detailed segmentation rules:

**TTS Segmentation Rules**

*Rule 1*: Natural punctuation boundaries — e.g., periods, commas, semicolons
*Rule 2*: Discourse markers and conjunctions — e.g., "however", "therefore", "and"
*Rule 3*: Syntactic boundaries — e.g., subject-predicate splits, subordinate clauses

Each segment typically contains 7-15 words to balance synthesis quality and parallelization efficiency. This granularity ensures both semantic completeness and natural prosody while maintaining computational efficiency. For example, the sentence "The cat sat on the mat, while the dog slept peacefully" would be split into two segments: "The cat sat on the mat" and "while the dog slept peacefully".

**Commit rules.** Canceled TTS segments are not written into memory. For audio inputs, only finalized ASR transcripts are committed for reuse; partial hypotheses remain ephemeral.

## E. LLM Orchestration Algorithm

This section provides the full orchestration pipeline used throughout our experiments. The controller emits a *tokens-only* plan under a closed vocabulary; the router deterministically validates and dispatches mapping-bound actions with cache-or-call semantics; a text-only generator produces the final response conditioned on retrieved evidence. Modality-specific preprocessing (ASR/OCR/frame extraction) and rendering (TTS) are standard modules and are detailed in Appendix D.

---

**Algorithm 1** Training-free LLM Orchestration (Controller–Router–Generator)

---

1: **Input:** Multimodal user input $u_t$, text query $q_t$, compact memory view $\widehat{M}_{t-1}$, full memory pool $M_{t-1}$, expert mapping $\mathcal{E}$
2: **Output:** Text response $o_t$ (optionally synthesized to audio by TTS)
3: **procedure** ORCHESTRATE($u_t, q_t, \widehat{M}_{t-1}, M_{t-1}, \mathcal{E}$)
4:    $S_t \leftarrow f_{\text{ctrl}}(q_t, \widehat{M}_{t-1})$        // controller emits *control tokens only*
5:    **if** $\neg$ ISVALIDTOKENSONLY($S_t$) **then**
6:       $S_t \leftarrow$ SAFEDEFAULTROUTE()       // no external expert calls
7:    **end if**
8:    **if** [S.stop] $\in S_t$ **then return** *null*       // interruption handled by interaction manager
9:    **if** [S.listen] $\in S_t$ **then return** *null*       // await additional input
10:    $\delta(S_t) \leftarrow$ VALIDATEANDPLAN($S_t, \mathcal{E}$)       // deterministic parsing + mapping checks
11:    $D_t \leftarrow$ CACHEORCALL($\delta(S_t), u_t, q_t, M_{t-1}$)       // hit-or-miss per action
12:    $z_t \leftarrow$ COMPOSE($q_t, \widehat{M}_{t-1}, D_t$)       // text-only evidence context
13:    $o_t \leftarrow f_{\text{gen}}(z_t)$       // generator emits *text only*
14:    $M_t \leftarrow$ COMMIT($M_{t-1}, q_t, D_t, o_t$)       // commit only completed outputs
15:    $\widehat{M}_t \leftarrow h_{\text{compress}}(M_t)$       // bounded memory view for next turn
16:    **return** $o_t$
17: **end procedure**
18: **function** VALIDATEANDPLAN($S_t, \mathcal{E}$)
19:    parse $S_t$ and keep routing tokens $s_k \in \mathcal{S}_{\text{route}}$
20:    map each routing token to a mapping-bound action $a_k = \Gamma(s_k)$
21:    **return** ordered plan $\delta(S_t) = [a_1, \ldots, a_{K'}]$
22: **end function**
23: **function** CACHEORCALL($\delta(S_t), u_t, q_t, M_{t-1}$)
24:    $D_t \leftarrow []$
25: **for** $a_k$ in $\delta(S_t)$ **do**
26:    (hit, $d$) $\leftarrow$ RETRIEVE($a_k, u_t, q_t, M_{t-1}$)
27:    **if** $\neg$ hit **then** $d \leftarrow$ RUNEXPERT($\pi(a_k), a_k, u_t$)       // bounded by runtime policy
28:    append $d$ to $D_t$
29: **end for**
30:    **return** $D_t$
31: **end function**

---

**Multi-turn Full-Duplex Workflow.** We include the full multi-turn interaction loop used in the main paper.

---

**Algorithm 2** Multi-turn Full-Duplex Workflow

---

**Require:** Mapping $\mathcal{E}$ with policies (timeout, cancellable); protocol vocabulary $\mathcal{S}$
1: $M \leftarrow \emptyset; \hat{M} \leftarrow \emptyset$; playback buffer $\mathcal{B} \leftarrow \emptyset$; async jobs $\mathcal{J} \leftarrow \emptyset$
2: **while** session active **do**
3:    $\tilde{u} \leftarrow$ STAGE1($u$); $q \leftarrow$ SELECTQUERY($\tilde{u}$); $\mathcal{K} \leftarrow$ KEYS($\tilde{u}$)
4:    **if** BARGEINDETECTED($\tilde{u}$) **and** $\mathcal{B} \neq \emptyset$ **then**
5:       CANCELINFLIGHTJOBS($\mathcal{J}, \mathcal{E}$); $\mathcal{B} \leftarrow \emptyset$
6:    **end if**
7:    $S \leftarrow f_{\text{ctrl}}(q, \hat{M})$
8:    $\delta \leftarrow$ VALIDATEANDPLAN($S; \mathcal{E}, \mathcal{K}$)
9:    **parallel for** $a \in \delta$ **do** $d(a) \leftarrow$ CACHEORCALL($a, \tilde{u}, M; \mathcal{E}$); ENQUEUE($\mathcal{J}$, job($a$)) **end for**
10:    $D \leftarrow [d(a) \mid a \in \delta]$; $o^{\text{text}} \leftarrow f_{\text{gen}}($COMPOSE($q, \hat{M}, D$))
11:    $M \leftarrow$ COMMIT($M, D, o^{\text{text}}$); $\hat{M} \leftarrow$ COMPRESS($M$)
12:    **if** SPEAKENABLED($S$) **then**
13:       $\{c_i\}_{i=1}^{B} \leftarrow$ CHUNK($o^{\text{text}}; \mathcal{P}, L$)
14:       **parallel for** $i = 1$ to $B$ **do** $J_i \leftarrow$ ASYNCTTS($c_i$); ENQUEUE($\mathcal{J}, J_i$) **end for**
15:       **for** $i = 1$ to $B$ **do**
16:          **if** STOPFLAGRAISED() **or** BARGEINDETECTED(STAGE1(u)) **then**
17:             CANCELPENDINGTTS($\{J_j\}_{j \geq i}$); **break**
18:          **end if**
19:          $y_i \leftarrow$ AWAIT($J_i$); STREAMPLAY($y_i$); update $\mathcal{B}$
20:       **end for**
21:    **else**
22:       EMITTEXT($o^{\text{text}}$)
23:    **end if**
24: **end while**

---

# F. Additional Experimental Results

## F.1. Parameter-Matched Routing Analysis

To separate the benefit of orchestration from raw expert scale, we evaluate routed configurations with different average active visual-parameter budgets on Video-MME. The controller is fixed, while the visual expert pool contains Qwen2.5-VL-7B, Qwen2.5-VL-32B, and Qwen2.5-VL-72B. Each routed variant invokes a mixture of experts; therefore, *Avg. Visual Expert Params* denotes the weighted average parameter count of invoked visual experts rather than the maximum available expert size. The main table reports the best routed setting as 53.8B average active visual expert parameters; the 14B controller is fixed across routed variants.

**Table 4.** Parameter-matched Video-MME analysis under a fixed 14B controller. Routed variants differ only in the distribution over the visual expert pool.

| Model | Avg. (B) | Acc. (%) | 7B (%) | 32B (%) | 72B (%) |
|---|---|---|---|---|---|
| Qwen2.5-VL | 72.0 | 65.74 | – | – | – |
| LLaVA-OV | 72.0 | 66.20 | – | – | – |
| Ours (Qwen2.5-VL) | 19.7 | 62.07 | 49.20 | 50.80 | 0.00 |
| Ours (Qwen2.5-VL) | 35.2 | 62.99 | 56.60 | 0.00 | 43.40 |
| Ours (Qwen2.5-VL) | 45.6 | 64.62 | 20.10 | 33.30 | 46.60 |
| Ours (Qwen2.5-VL) | 53.8 | 65.58 | 0.00 | 45.60 | 54.40 |

The 53.8B average-active-parameter variant reaches 65.58%, close to dense 72B baselines, while avoiding uniform reliance on the largest expert. Across routed settings, the controller assigns simple cases to lighter experts and reserves larger experts for more difficult cases. This supports the interpretation that the gain comes from task-adaptive expert selection, with orchestration overhead remaining below 12% in the latency analysis.

## F.2. Expert-Usage Statistics by Benchmark

Table 5 reports how often the router selects the 32B and 72B Qwen2.5-VL experts across benchmarks. Easier general multimodal benchmarks are routed more often to the 32B expert, whereas more demanding video and holistic settings trigger a more balanced distribution. This provides a direct view of the computation allocation underlying the end-to-end scores.

**Table 5.** Expert-selection frequency across benchmarks. Values report the percentage of routed visual expert calls assigned to each Qwen2.5-VL expert.

| Benchmark | Qwen2.5-VL-32B (%) | Qwen2.5-VL-72B (%) |
|---|---|---|
| MME | 73.10 | 26.90 |
| MMBench-EN | 69.80 | 30.20 |
| MMStar | 71.80 | 28.20 |
| LVBench | 52.40 | 47.60 |
| MMMU | 63.60 | 36.40 |
| Video-MME | 51.30 | 48.70 |
| WorldSense | 54.20 | 46.80 |

## F.3. Direct Router Reliability Evaluation

We quantify whether the controller follows the closed-vocabulary routing schema by measuring valid control-token generation on held-out benchmark queries. We compare the training-free controller against a supervised router baseline obtained by LoRA fine-tuning Qwen2.5-14B on 7,500 successful multi-turn routing trajectories, with 1,200 held-out examples split by independent interaction sessions. The baseline uses LoRA rank 16,

alpha 32, dropout 0.05, learning rate $2 \times 10^{-4}$, global batch size 256, and a cosine scheduler; epoch 3 is selected from a 1–7 epoch sweep.

**Table 6.** Closed-vocabulary routing success. Success means that the controller emits a valid, parseable control-token sequence in a single pass.

| Controller | Video-MME | WorldSense | Success (%) |
|---|---|---|---|
| Supervised-trained | 2692/2700 | 3136/3172 | 99.20 |
| Training-free (ours) | 2679/2700 | 3144/3172 | 99.10 |

The small 0.1% gap indicates that the base controller's instruction-following ability is already sufficient for the routing protocol. Remaining invalid outputs are rejected by deterministic validation and fall back to a safe listen/text-only state, so invalid control strings do not execute expert calls.

## F.4. Interactive Multi-turn Evaluation

To better reflect real-time interaction, we restructure Video-MME questions into continuous three-round dialogue sessions per video. This setting tests shifting user goals, corrections, and memory reuse rather than isolated single-turn QA. Table 7 reports per-round accuracy, memory hit rate, and expert usage among cache misses.

**Table 7.** Interactive three-round Video-MME evaluation. Expert usage is measured only among cache misses, because cache hits reuse committed memory records without invoking a visual expert.

| Round | Acc. (%) | Memory Hit (%) | 32B Misses (%) | 72B Misses (%) |
|---|---|---|---|---|
| Round 1 | 66.70 | 0.00 | 51.30 | 48.70 |
| Round 2 | 69.20 | 96.20 | 71.20 | 28.80 |
| Round 3 | 64.30 | 88.30 | 60.30 | 39.70 |

Memory reuse is strongest in the second round, where follow-up questions often refer to evidence already acquired in the first turn. The third round remains high but lower, reflecting more frequent topic shifts and corrections.

## F.5. Evidence-Key Robustness and Failure Cases

**Evidence-key design.** The checksum component $h$ in Eq. 1 is a strict SHA-1 checksum computed at the segment level, not a perceptual hash. Continuous media is partitioned before key generation, for example by VAD intervals for audio and by chunk/frame identifiers for video. Consequently, a local perturbation creates a local cache miss for the affected segment, while unchanged segments remain reusable. We choose exact matching to avoid erroneous reuse from fuzzy semantic matching; the trade-off is bounded local recomputation under minor input variation.

**Component instantiation.** ASR requests use Whisper-Small. OCR and math requests are activated through routing tokens such as `[S.need_ocr]` and `[S.need_math]`; in our experiments these requests are served by the integrated multimodal expert pool consisting of Qwen2.5-VL-7B, Qwen2.5-VL-32B, and Qwen2.5-VL-72B.

**Failure modes.** The main residual failure case is routing ambiguity when a query lacks a clear modality trigger. Such cases are uncommon: invalid or unresolved routing occurs in 0.8% of Video-MME queries (21/2700) and 0.9% of WorldSense queries (28/3172). The runtime handles them with a safe `[S.listen]`

or text-only fallback, which may increase latency or produce a conservative no-op response but avoids invalid expert execution. Other practical limits include reduced responsiveness under frequent consecutive interruptions and degraded fine-grained temporal counting when event frequency exceeds the sparse frame-sampling rate.

**Latency and barge-in behavior.** In the current unoptimized prototype, end-to-end throughput reaches 944.81 output tokens/s on a single NVIDIA H100 80GB GPU for queries with an average context length of about 512 tokens. The orchestration layer adds less than 12% latency overhead, and retry rate remains below 1.2%. During barge-in events, immediate cancellation prevents more than 90% of scheduled compute from being spent on obsolete output or pending expert/TTS work.

**WorldSense Task Breakdown.** Table 8 reports task-type breakdown on WorldSense, highlighting the performance gap between semantic/causal reasoning and fine-grained temporal tracking.

**Table 8.** WorldSense task-specific breakdown showing performance variation across semantic reasoning vs. temporal tracking tasks. Overall accuracy is 44.07% (Table 1).

| Task Type | Acc (%) | Correct/Total |
|---|---|---|
| *High-Performance (Semantic & Causal)* | | |
| Emotion Change | 60.42 | 58/96 |
| Hallucination Detection | 60.00 | 54/90 |
| Temporal Prediction | 58.18 | 64/110 |
| Causal Reasoning | 57.62 | 87/151 |
| *Audio-Related (Omni-Modal)* | | |
| Audio Source Localization | 42.50 | 51/120 |
| Audio Recognition | 42.24 | 49/116 |
| Audio Counting | 34.44 | 31/90 |
| *Challenging (Temporal Tracking)* | | |
| Spatial Relation | 33.50 | 66/197 |
| Temporal Localization | 33.14 | 56/169 |
| Object Counting | 31.22 | 64/205 |
| Action Counting | 23.03 | 38/165 |

