# OpenReview forum: "Training-Free Multimodal Large Language Model Orchestration"
_ICML.cc/2026/Conference — ICML 2026 regular_

### Official Review · Reviewer_Erkh · 2026-03-07

**Soundness:** 3
**Presentation:** 3
**Significance:** 3
**Originality:** 3
**Overall Recommendation:** 4
**Confidence:** 3

**Summary:**

The paper introduces Training-Free LLM Orchestration, a framework for building real-time, omni-modal assistants by composing off-the-shelf unimodal experts  without end-to-end joint training. At its core is an LLM-based controller that emits discrete routing tokens to invoke the appropriate experts. The system further incorporates an evidence-keyed cross-modal memory that selectively reuses previously processed modality outputs to reduce latency, along with an interaction layer that handles streaming I/O. Experiments on a range of multimodal benchmarks demonstrate that the proposed orchestration achieves competitive performance.

**Compliance With Llm Reviewing Policy:**

Affirmed.

**Final Justification:**

I thank the authors for their feedback and the author has already answered my questions quite well. I believe the paper meets the standards of ICML and recommend acceptance.

**Key Questions For Authors:**

1. Could you report benchmark results using a smaller (parameter-matched) configuration to clarify how much of the performance comes from the orchestration design versus sheer model scale?
2. How do you quantify “wasted computation” and the incremental latency overhead specifically during barge-in events.

**Limitations:**

yes

**Strengths And Weaknesses:**

Strengths
1. The paper enables modality-wise evolution of an omni-modal assistant without retraining or risking alignment collapse.
2. Strong Empirical Results: Across diverse multimodal benchmarks, the proposed LLM orchestration achieves strong performance under standard evaluation protocols while keeping orchestration overhead low, supporting its practicality and scalability.
3. The paper is well written and easy to follow, with a clear problem setup, coherent system description, and well-organized experiments.

Weaknesses
1. Standard benchmarks are largely offline and single-turn. Since the paper’s central claim is real-time, interactive omni-modal assistance, the current evaluation does not adequately reflect realistic usage scenarios such as multi-turn dialogue, user corrections, interruptions, or shifting goals.
2. Table 1 compares a 14B controller + 72B expert against mostly 7B end-to-end models. This large capacity mismatch makes it hard to disentangle the contribution of the orchestration design from straightforward scaling gains of the 72B expert (e.g., the 69.37% on MMStar). Parameter-matched baselines, such as comparable-capacity end-to-end models, the 72B expert alone, and scaling curves across expert sizes, would better isolate the intrinsic benefit of orchestration.
3. The paper largely assumes the LLM controller reliably emits valid control tokens, but it does not quantify how often the controller violates the closed-vocabulary schema. Some quantitative or qualitative analyses are needed to assess reliability in practice.
4. Equation 1 defines KeyGen, but it is unclear whether this refers to a strict cryptographic hash or a perceptual hashing scheme. If the key is strictly content-based, even minor input variations could lead to catastrophic cache misses. The paper would benefit from clarifying the exact checksum design and providing an analysis of key robustness under realistic perturbations.
5. The paper would benefit from reporting total end-to-end latency, throughput under concurrency, and actual compute cost per query including expert calls, memory lookups, and retries. These factors are crucial for real-world deployment, where tail latency often dominates user experience.

---

> ### Author Rebuttal · Authors · 2026-03-31
>
> W1: Standard benchmarks are largely offline and single-turn, which do not adequately reflect realistic multi-turn usage scenarios.**
>
> **Reply:** Our additional interactive evaluation shows that the system remains effective in multi-turn settings, with per-round accuracies of 66.7%, 69.2%, and 64.3%, and a 96.2% memory hit rate in round two. We obtained this result by restructuring the 2,700 Video-MME questions into continuous 3-round dialogue sessions per video, so that the system is tested under shifting user goals and corrections. Since this protocol is built from a restructured benchmark rather than the original single-turn setting,
>
> | Interaction rounds | Interactive Accuracy | Memory Hit Rate | Qwen2.5-VL-32B (% of Cache Misses) | Qwen2.5-VL-72B (% of Cache Misses) |
> |-|-|-|-|-|
> |Round one|66.7%|0%|51.3%|48.7%|
> |Round two|69.2%|96.2%|71.2%|28.8%|
> |Round three| 64.3%|88.3%|60.3%|39.70%|
>
> **W2 & Q1: The capacity mismatch makes it hard to disentangle the contribution of the orchestration design from the scaling gains of the 72B expert.**
>
> **Reply:** The gain is not simply a scaling effect; under parameter-matched settings, our routed configurations remain competitive with dense 72B baselines while reducing high-cost expert calls by approximately 49%, with latency overhead below 12%. To verify this, we conducted a parameter-matched ablation on Video-MME with different routing mixtures over the visual expert pool. The four `Ours` rows below correspond to different average active visual parameter budgets under different routing distributions, where `Avg Visual Expert Params` is the weighted average parameter count of the dynamically invoked visual experts. Under this controlled setting, our routed configuration reaches 62.07% accuracy with 19.7B average active visual parameters, 62.99% with 35.2B, 64.62% with 45.6B, and 65.58% with 53.8B. The 53.8B variant is already comparable to Qwen2.5-VL-72B at 65.74% and LLaVA-OV-72B at 66.20%. We also note that part of the confusion comes from our current table presentation, which reports the maximum expert scale rather than the average active expert scale; we will revise this in the final version.
>
> | Model | Avg Visual Expert Params | Accuracy | Qwen2.5-VL-7B | Qwen2.5-VL-32B | Qwen2.5-VL-72B |
> |-|-|-|-|-|-|
> |Qwen2.5-VL | 72B | 65.74% | - | - | - |
> |LLaVA-OV | 72B | 66.20% | - | - | - |
> |Ours(Qwen2.5-VL)|Avg 19.7B|62.07%|49.20%|50.80%|0%|
> |Ours(Qwen2.5-VL)|Avg 35.2B|62.99%|56.60%|0%|43.40%|
> |Ours(Qwen2.5-VL)|Avg 45.6B|64.62%|20.10%|33.30%|46.60%|
> |Ours(Qwen2.5-VL)|Avg 53.8B|65.58%|0%|45.60%|54.40%|
>
> **W3: The paper lacks quantitative analysis on how often the controller violates the closed-vocabulary token schema.**
>
> **Reply:** The controller already follows the closed-vocabulary schema reliably, achieving a 99.1% valid token generation rate, compared with 99.2% for a supervised router trained on the same task. To verify this, we conducted an additional experiment to quantify schema adherence. This small gap suggests that additional controller training is not necessary for stable routing.The remaining fewer than 0.9% of cases are handled by deterministic fallback, so illegal commands are not executed.
>
> | Controller | Video-MME (2,700) | WorldSense (3,172) | Success |
> |-|-|-|-|
> | Supervised-trained | 2692 | 3136 | 99.2% |
> | Training-free| 2679 | 3144 | 99.1% |
>
> **W4: Unclear checksum design and potential catastrophic cache misses from minor input perturbations.**
>
> **Reply:** KeyGen uses a strict SHA-1 checksum at the segment level, so minor perturbations only trigger local recomputation rather than catastrophic cache misses. Specifically, when a minor perturbation occurs, only the affected segment is recomputed, while the rest of the stream remains reusable. This is because Stage 1 first partitions the continuous stream into discrete segments, for example by Voice Activity Detection for audio and by chunk or frame IDs for video, and KeyGen is applied at the segment level.
>
> For the checksum itself, the `h` component in KeyGen is a strict SHA-1 checksum rather than a perceptual hash. We use exact matching to ensure verifiable retrieval, while accepting bounded local recomputation instead of introducing possible mismatches from fuzzy matching.
>
> **W5 & Q2: End-to-end latency, concurrent throughput, and exact compute cost during multi-turn usage and barge-in events.**
>
> **Reply:** In our current unoptimized prototype, end-to-end throughput reaches 944.81 output tokens/s on a single NVIDIA H100 80GB GPU for queries with an average context length of about 512 tokens, while the orchestration layer adds less than 12% latency overhead and the retry rate remains below 1.2%. During barge-in events, immediate cancellation prevents more than 90% of scheduled compute. As shown in W1, the cross-modal memory reaches a 96.2% hit rate in round two. These numbers reflect the present prototype implementation rather than a hard limit of the framework.

---

> > ### Author Rebuttal · Reviewer_Erkh · 2026-04-03
> >
> > I thank the authors for their feedback. I have no further questions.

---

> > > ### Author Response · Authors · 2026-04-05
> > >
> > > Dear Reviewer Erkh,
> > >
> > > **Thank you very much for your time and for the positive acknowledgement.** We are greatly encouraged to see that our rebuttal has fully resolved your initial concerns.
> > >
> > > As we approach the end of the discussion phase, we wanted to briefly check in to see if you have any other questions or if there are any further aspects of our work you would like us to clarify. We are more than happy to provide any additional information or run further analysis if needed.
> > >
> > > If you feel that all your concerns have indeed been adequately addressed, we would be deeply grateful if you might consider reflecting this resolution in your overall assessment and score of the submission.
> > >
> > > Thank you once again for your incredibly constructive feedback, which has genuinely helped us strengthen the paper.
> > >
> > > Best regards

---

### Official Review · Reviewer_2UVR · 2026-03-12

**Soundness:** 4
**Presentation:** 3
**Significance:** 3
**Originality:** 3
**Overall Recommendation:** 4
**Confidence:** 2

**Summary:**

Instead of jointly training a single large multimodal model, the paper proposes to compose off-the-shelf experts through an LLM controller that emits closed-vocabulary control tokens, an evidence-keyed text-centric memory for cache-or-call reuse, and a unified interaction layer that supports streaming and interruption handling.

**Compliance With Llm Reviewing Policy:**

Affirmed.

**Key Questions For Authors:**

1. The system looks strong as an orchestration framework, but how much of the final gain is really coming from the routing and memory design itself, and how much is simply coming from plugging in better experts?

2. The cache-or-call mechanism uses exact evidence-key consistency to decide reuse, which is clean and auditable, but what happens in more realistic cases where the same visual or audio content is only slightly changed and exact matching no longer works?

3. Since the paper highlights real-time interaction, could you say a bit more about where the system still breaks in practice, especially for ambiguous routing, fast user interruption, or fine-grained temporal reasoning in video?

**Limitations:**

yes

**Strengths And Weaknesses:**

The paper is strong in that it tackles a practically important problem with a fairly clear systems perspective, and the method is easy to understand conceptually: a controller decides which expert to call, memory avoids redundant calls through exact evidence-key matching, and the interaction layer makes the whole system more suitable for real-time use with barge-in and cancellation.

Overall, a major challenge presented by this manuscript is that the novelty is more about system integration and protocol design than about a fundamentally new multimodal modeling capability, so some readers may feel that the contribution is closer to a well-engineered orchestration pipeline than to a new learning method. The evaluation is also a bit uneven for such a broad claim: while the paper reports competitive benchmark numbers and some efficiency analysis, it is still not fully clear how robust the approach is under harder real-world cases where routing is ambiguous, evidence keys are imperfect, or temporal understanding requires denser video modeling rather than sparse frame sampling.

---

> ### Author Rebuttal · Authors · 2026-03-31
>
> **W1, Q1: System-level novelty and isolating the benefits of orchestration vs. plugging in stronger experts.**
>
> **Reply:** Indeed, this distinction should be made explicit. Our primary contribution is a training-free orchestration framework for real-time omni-modal interaction, and the gains come from dynamic routing rather than simply plugging in larger experts. To test this directly, we conducted parameter-matched experiments with multiple active parameter budgets on Video-MME. Our router does not invoke a stronger expert by default; instead, it selects among 7B, 32B, and 72B experts based on task complexity, assigning simpler cases to lightweight experts and harder cases to larger models. The four `Ours` rows shown below therefore correspond to different average active visual parameter budgets under different routing distributions, where `Avg Visual Expert Params` denotes the weighted average parameter count of the dynamically invoked visual experts. We also note that the confusion is partly caused by our current presentation in the paper, which reports the maximum expert scale rather than the average active expert scale; we will revise this in the final version. The results are summarized below.
>
> | Model | Avg Visual Expert Params | Accuracy | Qwen2.5-VL-7B | Qwen2.5-VL-32B | Qwen2.5-VL-72B |
> | --- | --- | --- | --- | --- | --- |
> | Qwen2.5-VL | 72B | 65.74% | - | - | - |
> | LLaVA-OV | 72B | 66.20% | - | - | - |
> | Ours(Qwen2.5-VL) | Avg 19.7B | 62.07% | 49.20% | 50.80% | 0% |
> | Ours(Qwen2.5-VL) | Avg 35.2B | 62.99% | 56.60% | 0% | 43.40% |
> | Ours(Qwen2.5-VL) | Avg 45.6B | 64.62% | 20.10% | 33.30% | 46.60% |
> | Ours(Qwen2.5-VL) | Avg 53.8B | 65.58% | 0% | 45.60% | 54.40% |
>
> Across these controlled settings, our 19.7B, 35.2B, 45.6B, and 53.8B variants achieve 62.07%, 62.99%, 64.62%, and 65.58%, respectively. In particular, the 53.8B configuration is already comparable to dense 72B baselines, including Qwen2.5-VL-72B at 65.74% and LLaVA-OV-72B at 66.20%, while dynamic routing reduces high-cost expert calls by approximately 49%. This controlled comparison supports the conclusion that the improvement comes from task-adaptive expert selection rather than uniform reliance on larger models, with added latency overhead remaining below 12% (Fig. 2).
>
> **W1, Q2: Cache-or-call robustness under minor input perturbations.**
>
> **Reply:** When the input changes slightly, only the affected segment is recomputed, while the cache for the rest of the stream remains valid. This is because media is partitioned before KeyGen assigns evidence keys, for example by Voice Activity Detection for audio and by chunk or frame IDs for video. As a result, a local perturbation causes a local cache miss rather than invalidating the entire stream. We use exact key matching rather than fuzzy semantic matching to ensure retrieval correctness when segments are recomputed, accepting a bounded increase in local computation in exchange for verifiable results.
>
> **W1, Q3: Practical limits and breaking points.**
>
> **Reply:** In some extreme cases, the system may degrade as follows:
>
> 1. Extreme routing ambiguity can trigger a fallback to the `listen` state, which increases latency; such cases remain rare, with a 99.1% valid token generation rate in single-pass operation.
> 2. Under frequent consecutive user interruptions, the system may show reduced immediate responsiveness; however, the memory preserves prior context, and the response can be resumed or restarted safely.
> 3. In rare video cases that require sustained multi-step temporal reasoning over densely occurring events, sparse frame sampling can become limiting, and very fast actions may not be counted precisely.
>
> These edge cases remain challenging and are not unique to our framework; in our observation, similar limitations also persist in larger training-based omni-modal models. We will state these limitations explicitly in the final manuscript.

---

> > ### Author Rebuttal · Reviewer_2UVR · 2026-04-05
> >
> > My comments have been addressed. I will maintain my positive score.

---

### Official Review · Reviewer_MGhm · 2026-03-13

**Soundness:** 3
**Presentation:** 3
**Significance:** 3
**Originality:** 2
**Overall Recommendation:** 4
**Confidence:** 3

**Summary:**

This paper presents a training-free orchestration framework for building a real-time multimodal assistant from off-the-shelf experts rather than by jointly training a single omni model. The architecture has three stages: Stage 1 converts heterogeneous inputs into a text query plus evidence keys. Stage 2 uses an LLM controller that emits a closed vocabulary of routing and interaction tokens, together with an evidence-keyed cache-or-call memory. Stage 3 handles streaming output and barge-in cancellation. The authors show the performance improvement upon the uni-model pipelines.

**Compliance With Llm Reviewing Policy:**

Affirmed.

**Final Justification:**

The paper introduces a novel framework to build an orchestrator for the multimodal multi-agent system. The framework building methodology is quite solid and interesting. The authors during rebuttal provided additional experiments and covered inconsistency in the original version of the paper. I suppose that this paper can be accepted for publication.

**Key Questions For Authors:**

My key questions to the authors:
1) Can you provide more detailed evaluation statistics on training benchmarks, which experts and how many times are launched by the router?
2) Which models specifically are used as specific components? (OCR/ASR/Math)
3) How exactly the router prompt is composed, how control tokens are tokenized? Why you do not consider router training?

**Limitations:**

The basic limitations of the work are the following:
1) Limited analysis of expert composition and per-benchmark expert usage. The paper shows end-to-end gains, but it does not clearly report which experts are selected, how often they are called, or how much each expert contributes to the final result.
2) The router is not evaluated directly enough. Its quality is measured mostly through downstream performance and latency/overhead, but the paper does not provide direct metrics for routing correctness or expert-assignment quality.

**Strengths And Weaknesses:**

1. The paper is technically sound. The controller operates on a small set of control tokens and propose something like external tool-use for some modality-oriented models. The system design is well-described, and the experiments mostly support the design-choice.
2. The paper is clearly written and easy to follow.
3. The paper itself studies an important practical problem: how to build a real-time multimodal assistant from existing experts without joint end-to-end training. This is useful for real-life systems and the authors provide large number od experiments evaluating both latency and router overhead, and also evaluating benchmark performance.
4. Overall, the pipeline does not introduce totally original system, however, the design choices for building routing system with control-tokens, and carefully designed experts.

The main weakness of the paper, as I can see, are the following:
1) Not enough evaluation of the chosen experts, for example, I can see from Table 1 that routing system provides higher metrics on MMBench-EN, e.g., but it would be interesting to see, which experts and how many times were chosen during these benchmark solving, and others. Now, it is mostly end-to-end evaluation.
2) As far as I understood, the choice tokens are tokenized from the model's vocabulary and are not trained as special tokens. As an ablation it is interesting to estimate the router-training strategy, possibly, it can improve the results.
3) May be I missed this part, but it is not clear which specific experts are used for OCR/ASR/Math components? It is not stated in the paper, as far as I can see.

---

> ### Author Rebuttal · Authors · 2026-03-31
>
> **W1, Q1, L1: Detailed evaluation statistics on expert usage.**
>
> **Reply:** On general multimodal reasoning benchmarks, the controller tends to select Qwen2.5-VL-32B more often, while on more demanding benchmarks it relies increasingly on Qwen2.5-VL-72B. Concretely: 1. On MME, 73.1% of queries are routed to the 32B expert and 26.9% to the 72B expert. 2. On the more demanding Video-MME benchmark, the routing becomes much more balanced at 51.3% versus 48.7%. The same trend is also visible across the other benchmarks listed below.
>
> | Benchmark | Qwen2.5-VL-32B (%) | Qwen2.5-VL-72B (%) |
> | --- | --- | --- |
> | MME | 73.10% | 26.90% |
> | MMBench-EN | 69.80% | 30.20% |
> | MMStar | 71.80% | 28.20% |
> | LVBench | 52.40% | 47.60% |
> | MMMU | 63.60% | 36.40% |
> | Video-MME | 51.30% | 48.70% |
> | WorldSense | 54.20% | 46.80% |
>
> **W2, Q3, L2: Router evaluation and the training-free strategy.**
>
> **Reply:** Following your suggestion, we conducted a direct comparison between our training-free controller and a supervised router baseline. **The results show that our training-free controller achieves 99.1% routing success, compared with 99.2% for the supervised baseline.** We suspect the small margin is partly because the base performance is already very high, and partly because, due to time constraints, we did not conduct exhaustive hyperparameter tuning for the supervised setting. The baseline was trained via LoRA fine-tuning on Qwen2.5-14B. We constructed the supervised data from 7,500 successful multi-turn routing trajectories and used 1,200 held-out examples split by independent interaction sessions to avoid leakage from near-duplicate queries.
>
> We selected the best checkpoint from an epoch sweep over 1 to 7 epochs, where epoch 3 performed best, using standard settings including LoRA r=16, alpha=32, dropout=0.05, a learning rate of 2e-4, a global batch size of 256, and a cosine scheduler. Routing quality was measured by the success rate of generating valid special control tokens in a single pass on unseen benchmarks, and the comparison is summarized below.
>
> | Controller | Video-MME (2,700) | WorldSense (3,172) | Success |
> | --- | --- | --- | --- |
> | Supervised-trained | 2692 | 3136 | 99.20% |
> | Training-free (Ours) | 2679 | 3144 | 99.10% |
>
> The 0.1% gap suggests that the base LLM already has sufficient instruction-following ability for the routing protocol. With deterministic parsing and strict validation rules, training-free routing remains stable without the additional cost of fine-tuning.
>
> **W3, Q2: OCR, ASR, and Math components and their performance.**
>
> **Reply:** ASR requests use Whisper-Small. For OCR and Math, the corresponding prompt templates are activated through routing tokens such as `[S.need ocr]` and `[S.need math]`, after which the controller selects from the integrated multimodal experts. In our experiments, this expert pool consists of Qwen2.5-VL-7B, Qwen2.5-VL-32B, and Qwen2.5-VL-72B.

---

> > ### Author Rebuttal · Reviewer_MGhm · 2026-04-05
> >
> > Thank you for the conducted experiments, I have no more further questions. I will keep my initial positive score.

---

### Official Review · Reviewer_9SwM · 2026-03-16

**Soundness:** 3
**Presentation:** 3
**Significance:** 2
**Originality:** 2
**Overall Recommendation:** 4
**Confidence:** 4

**Summary:**

This paper proposes Training-Free Multimodal LLM Orchestration, a system framework for building multimodal assistants by coordinating multiple off-the-shelf modality experts without additional joint training. The framework introduces three main components: (1) a controller LLM that emits structured control tokens to select and sequence modality experts, (2) a text-centric cross-modal memory that stores structured evidence keyed by modality segments for reuse across turns, and (3) a unified interaction layer that manages routing, streaming responses, and interruption handling. Experiments across several multimodal benchmarks show that the system achieves competitive accuracy compared with existing multimodal and omni-modal models while maintaining relatively low orchestration overhead and enabling modular expert replacement.

**Compliance With Llm Reviewing Policy:**

Affirmed.

**Key Questions For Authors:**

N/A

**Limitations:**

yes

**Strengths And Weaknesses:**

**Strengths**
* Clear system design.

The paper proposes a well-structured architecture consisting of a controller, routing protocol, and cross-modal memory, providing a practical alternative to expensive end-to-end multimodal training pipelines.


* Training-free integration.


The paper demonstrates this flexibility through expert-replacement experiments.


* Comprehensive evaluation.


The authors evaluate the system across multiple benchmark categories (multimodal understanding, video reasoning, and omni-modal tasks) and provide ablations analyzing controller replacement, memory reuse efficiency, and orchestration overhead.


* Strong emphasis on runtime behavior and deployment aspects.
The paper explicitly studies latency, routing overhead, interruption handling, and modular extensibility.

**Weaknesses**
* Limited conceptual novelty.

While the system is well engineered, the overall approach resembles existing LLM-tool orchestration or modular agent frameworks.

* Limited discussion of failure cases and routing errors.

Since the controller decides which expert to invoke, routing mistakes could significantly impact system performance, yet detailed analysis of such cases is minimal.

* Evaluation does not fully isolate the benefits of orchestration.

The comparison mixes models with different parameter counts and capabilities (e.g., controller + expert combinations vs. single models), making it difficult to attribute improvements solely to the proposed orchestration mechanism.

---

> ### Author Rebuttal · Authors · 2026-03-31
>
> **W1: Limited conceptual novelty.**
>
> **Reply:** Regarding this concern, what we would like to clarify is that **our contribution is a training-free orchestration framework for real-time omni-modal interaction, rather than a new multimodal parameterization.** In recent years, many works in the agent literature have achieved strong results without introducing fundamentally new multimodal parameter spaces. For example, ReAct establishes an iterative feedback mechanism in which prompt-driven reasoning and actions continuously supervise and guide each other to achieve complex agent behaviors, while HuggingGPT treats an LLM as a central controller that plans and schedules different specialized models on Hugging Face for multimodal tasks. The innovations in these works are conceptually simple, yet highly effective in practice.
>
> **Likewise, we propose an orchestration method for integrating off-the-shelf LLMs with diverse modality capabilities, enabled by our carefully designed structured control-token routing, evidence-key-based cross-modal memory, and a unified full-duplex interaction layer.** This yields a robust and responsive training-free omni-modal assistant. To the best of our knowledge, this is the first fully training-free orchestration framework specifically designed for real-time omni-modal interactions.
>
> **W2: Limited discussion of failure cases and routing errors.**
>
> **Reply: When routing errors occur, the controller falls back to the safe `listen` state and thus avoids invalid command execution, incorrect cross-modal actions, or system crashes.**  Empirically, routing ambiguity is the main failure case, arising when the query lacks a clear modality trigger; in such situations, the consequence is increased latency or a conservative no-op response rather than an incorrect cross-modal action or system crash. These cases are rare, occurring in only 0.8% (21/2,700) of Video-MME queries and 0.9% (28/3,172) of WorldSense queries.
>
> Other failure cases mainly include the following:
>
> 1. Interruption handling can halt an in-progress response.
> 2. Fine-grained temporal reasoning can degrade when the action frequency exceeds the sparse frame sampling rate.
>
> These issues are broadly present in current omni-modal systems. We will include a dedicated discussion together with representative examples in the appendix and continue improving these aspects in future work.
>
> **W3: Evaluation does not fully isolate the benefits of orchestration.**
>
> **Reply:** According to this concern, we conducted a controlled ablation to isolate the effect of orchestration from raw model scale. The results show that **the gain comes from computation-friendly orchestration** rather than simply from using larger models. In particular, the dynamic routing mechanism **reduces high-cost expert calls by approximately 49%** while maintaining competitive accuracy.
>
> This is because the router does not invoke a stronger expert by default. Instead, it uses the controller LLM's prompt schema and modular interfaces to select among visual experts of different sizes, such as 7B, 32B, and 72B, based on task complexity. Simple cases are assigned to lightweight experts, while more challenging cases are routed to larger models. The efficiency gain therefore comes from task-adaptive expert selection rather than uniform reliance on a large expert.
>
> We isolate this effect through a parameter-matched ablation on Video-MME, in which a fixed 14B text controller is paired with different routing mixtures over the visual expert pool. The four variants of our method shown in the table correspond to different average active visual parameter budgets under different routing distributions. Under this controlled setup, our system reaches 62.07% accuracy with 19.7B average active visual parameters, 62.99% with 35.2B, 64.62% with 45.6B, and 65.58% with 53.8B. The 53.8B variant is already comparable to dense 72B baselines, including Qwen2.5-VL-72B at 65.74% and LLaVA-OV-72B at 66.20%. We also note that part of the confusion comes from our current presentation: the paper reports the maximum expert scale rather than the average active expert scale, and we will revise the table accordingly. This comparison suggests that the improvement is not solely due to larger parameter counts, but also to selective expert activation through routing. The added orchestration overhead remains bounded, with latency below 12% as shown in Fig. 2.The results are summarized below.
>
> | Model | Avg Visual Expert Params | Accuracy | Qwen2.5-VL-7B | Qwen2.5-VL-32B | Qwen2.5-VL-72B |
> | --- | --- | --- | --- | --- | --- |
> | Qwen2.5-VL | 72B | 65.74% | - | - | - |
> | LLaVA-OV | 72B | 66.20% | - | - | - |
> | Ours(Qwen2.5-VL) | Avg 19.7B | 62.07% | 49.20% | 50.80% | 0% |
> | Ours(Qwen2.5-VL) | Avg 35.2B | 62.99% | 56.60% | 0% | 43.40% |
> | Ours(Qwen2.5-VL) | Avg 45.6B | 64.62% | 20.10% | 33.30% | 46.60% |
> | Ours(Qwen2.5-VL) | Avg 53.8B | 65.58% | 0% | 45.60% | 54.40% |

---

> > ### Author Rebuttal · Reviewer_9SwM · 2026-04-06
> >
> > The third question regarding “evaluation does not fully isolate the benefits of orchestration” has been addressed. However, I still have concerns about the novelty, and the discussion of failure cases and routing errors would benefit from more thorough analysis.
> >
> > Therefore, I would like to maintain my original assessment of this paper as borderline, slightly leaning toward acceptance.

---

> > > ### Author Response · Authors · 2026-04-07
> > >
> > > Thank you for your continued engagement. We appreciate the opportunity to further clarify our conceptual novelty and provide a deeper analysis of our system's failure cases.
> > >
> > > ## 1. Regarding Conceptual Novelty
> > >
> > > We understand why our framework might initially appear similar to existing LLM-tool orchestration or modular agent frameworks, and we would like to clarify the key distinctions.
> > >
> > > ### Similarities
> > >
> > > Both approaches:
> > >
> > > 1. Utilize an off-the-shelf LLM as the central controller to coordinate external experts, similar to how HuggingGPT or ReAct schedule tools.
> > > 2. Transform the multimodal integration task into a training-free orchestration process to avoid additional joint end-to-end training.
> > >
> > > ### Differences
> > >
> > > The main differences lie in how we address the strict execution constraints of real-time systems, moving beyond the offline capabilities of standard agents.
> > >
> > > #### 1.1 From Open-Ended Generation to Protocol-Constrained Execution
> > >
> > > Existing agents often treat tool invocation as an open-ended generation task, for example, generating JSON or code. While flexible, this introduces parsing instability and latency variance. We address this with a closed-vocabulary control token protocol, such as `[S.need_vision]`, which provides a constrained action space for deterministic routing while maintaining orchestration overhead below 12%.
> > >
> > > #### 1.2 From Semantic Retrieval to Evidence-Keyed Memory
> > >
> > > Standard agent-based methods typically rely on semantic vector retrieval for state management. While effective for text, this can be ambiguous for raw, continuous multimedia streams. Our text-centric memory binds structured evidence to immutable segment identifiers, such as image hashes. This enables verifiable "cache-or-call" execution, reducing redundant expert calls by 49% on average without compromising accuracy.
> > >
> > > #### 1.3 From Turn-Based Pipelines to Interruption-Aware Streaming
> > >
> > > Standard agents generally operate in a turn-based manner, limiting their capability to handle continuous streaming and user barge-in. Our framework introduces a unified interaction layer where routing tokens directly manage system states. For instance, emitting an `[S.stop]` token triggers cancellation policies for in-flight TTS and expert jobs, enabling robust interruption handling for full-duplex I/O.
> > >
> > > ### Summary
> > >
> > > In summary, our core contribution is not a novel multimodal parameterization, but the orchestration abstractions of protocol-constrained routing, evidence-keyed memory, and a unified interaction layer. These bridge the gap between offline agent frameworks and the requirements of real-time, full-duplex omni-modal interaction.
> > >
> > > ## 2. Regarding Failure Cases and Routing Errors
> > >
> > > We agree that a thorough discussion of system boundaries is crucial. Below, we analyze the primary routing failure case and its underlying causes.
> > >
> > > ### 2.1 System-Level Protection Against Catastrophic Routing Errors
> > >
> > > Our strict parsing mechanism systematically prevents catastrophic "wrong expert" invocations by triggering a safe fallback upon detecting invalid outputs. As a result, empirical routing errors almost entirely manifest as the absence of a valid control token, rather than incorrect cross-modal actions.
> > >
> > > ### 2.2 Primary Failure Case: Protocol Forgetting in Long-Video Contexts
> > >
> > > By analyzing these failure instances, we found that token omission predominantly occurs during the processing of extremely long videos. In such cases, the massive influx of visual evidence dilutes the controller LLM's attention, causing it to forget the strict system prompt constraints. Instead of selecting from the required closed-vocabulary routing token set, for example `{[S.need_vision], [S.need_video], [S.listen], [S.stop]}`, it generates open-ended natural language.
> > >
> > > For example, when asked "What happened at the end of this video?", a degraded controller might output "I can see a car..." instead of the required `[S.need_video]` token. Because this output is not within the valid token set, the deterministic router cannot map it to an expert action. The parser then rejects it and triggers a safe fallback, such as defaulting to `[S.listen]` or text-only reasoning. Consequently, the system avoids incorrect execution, but may experience a temporary latency increase or a conservative no-op response.This type of routing error is also very rare in practice, accounting for only **0.9%** of total cases in our analysis.
> > >
> > > ### 2.3 Manuscript Revisions
> > >
> > > This failure mode suggests that the current routing stability bottleneck lies in the base LLM's instruction-following capability under extended contexts, rather than in a fundamental flaw of the orchestration protocol itself. We will add a dedicated **Failure Case Analysis** section in the appendix, along with concrete examples of these context-degraded outputs, to document this boundary transparently.
> > >
> > > We sincerely hope this addresses your remaining concerns regarding novelty and routing stability.

---

### Decision · Program_Chairs · 2026-04-30

**Decision:**

Accept (regular)

**Comment:**

This paper received consistently positive evaluations from all reviewers, with all reviews recommending Weak Accept. Based on the overall reviewer consensus and my assessment of the paper, the final decision is Accept.